# High-Dose Nebulized Colistin Methanesulfonate and the Role in Hospital-Acquired Pneumonia Caused by Gram-Negative Bacteria with Difficult-to-Treat Resistance: A Review

**DOI:** 10.3390/microorganisms11061459

**Published:** 2023-05-31

**Authors:** Ilias Karaiskos, Aikaterini Gkoufa, Elena Polyzou, Georgios Schinas, Zoe Athanassa, Karolina Akinosoglou

**Affiliations:** 1First Department of Internal Medicine-Infectious Diseases, Hygeia General Hospital, 4, Erythrou Stavrou Str. & Kifisias, 15123 Athens, Greece; 2Infectious Diseases and COVID-19 Unit, Medical School, Laiko General Hospital, National and Kapodistrian University of Athens, 11527 Athens, Greece; katergouf@yahoo.gr; 3School of Medicine, University of Patras, 26504 Patras, Greece; polyzou.el@gmail.com (E.P.); georg.schinas@gmail.com (G.S.); akin@upatras.gr (K.A.); 4Department of Internal Medicine and Infectious Diseases, University General Hospital of Patras, 26504 Patras, Greece; 5Intensive Care Unit, Sismanoglio General Hospital, 15126 Athens, Greece; zathanassa@yahoo.gr

**Keywords:** colistin methanesulfonate, colistin, high-dose nebulized CMS, nebulizers, clinical efficacy, toxicity, nebulized 15 MIU CMS, aerolized, inhaled, high dose

## Abstract

Hospital-acquired pneumonia, including ventilator-associated pneumonia (VAP) due to difficult-to-treat-resistant (DTR) Gram-negative bacteria, contributes significantly to morbidity and mortality in ICUs. In the era of COVID-19, the incidences of secondary nosocomial pneumonia and the demand for invasive mechanical ventilation have increased dramatically with extremely high attributable mortality. Treatment options for DTR pathogens are limited. Therefore, an increased interest in high-dose nebulized colistin methanesulfonate (CMS), defined as a nebulized dose above 6 million IU (MIU), has come into sight. Herein, the authors present the available modern knowledge regarding high-dose nebulized CMS and current information on pharmacokinetics, clinical studies, and toxicity issues. A brief report on types of nebulizers is also analyzed. High-dose nebulized CMS was administrated as an adjunctive and substitutive strategy. High-dose nebulized CMS up to 15 MIU was attributed with a clinical outcome of 63%. High-dose nebulized CMS administration offers advantages in terms of efficacy against DTR Gram-negative bacteria, a favorable safety profile, and improved pharmacokinetics in the treatment of VAP. However, due to the heterogeneity of studies and small sample population, the apparent benefit in clinical outcomes must be proven in large-scale trials to lead to the optimal use of high-dose nebulized CMS.

## 1. Introduction

Hospital-acquired pneumonia (HAP), including ventilator-associated pneumonia (VAP), remains a challenging issue in critically ill patients in the hospital setting [1]. The incidence of VAP in the pre-COVID pandemic accounted for <15% of cases; however, VAP developed into a major issue during the pandemic era, with an overall crude incidence ranging from 40% to 60% [2,3]. In the era of COVID-19, the issue has become ever more pressing due to the increased occurrence of secondary nosocomial pneumonia in SARS-CoV-2-positive critically ill patients [4] and the unprecedented demand for invasive mechanical ventilation [5]. A worrisome issue is the predominance of Gram-negative etiology in the majority of HAP and VAP infections, with more concerning the high prevalence of multidrug-resistant organisms (MDR), as frequent as one-third of the hospital-acquired infections in patients with COVID-19 [6,7]. Carbapenem-based empiric antibiotic regimens have long been associated with improved outcomes and, thus, comprised the backbone of antibiotic therapy for this patient population [8]. However, alarming is the emergence of carbapenem-resistant Gram-negative bacteria (CR-GNB), which has raised global concern due to limited remaining treatment options and overall poor prognosis with existing regimens [9]. As such, new terminology has been used to describe these pathogens, specifically Gram-negative pathogens with difficult-to-treat resistance (DTR), practically defined as treatment-limiting resistance to all first-line agents, that is, all β-lactams, including carbapenems and β-lactamase inhibitor combinations, and fluoroquinolones [10].

On the other hand, newer β-lactam-β-lactamase inhibitors (i.e., ceftazidime-avibactam, meropenem-vaborbactam, and imipenem-cilastatin-avibactam) have been launched in the market for the treatment of DTR infections, including HAP and VAP infections. These agents are active against Enterobacterales producing extended-spectrum β-lactamase (ESBL), AmpC, and *Klebsiella pneumoniae* carbapenemase (KPC), whereas only avibactam inhibits certain class D β-lactamases, mainly OXA-48. The major drawback of these agents is that the β-lactam-β-lactamase inhibitors are ineffective against metallo-β-lactamases (MBL) as well as *Acinetobacter baumannii* [11].

Therefore, an older drug, i.e., intravenous (IV) colistin methanesulfonate (CMS), a pro-drug of colistin, is still considered a useful antimicrobial agent for the treatment of DTR infections, as it is active against carbapenem-resistant pathogens [12]. Nebulized CMS for the treatment of VAP caused by extensively drug-resistant (XDR) Gram-negative bacteria has also been applied, taking under consideration the theoretic advantages of achieving high drug concentrations at the infection site, considerably more than the minimal inhibitory concentration (MIC) of most causative microorganisms with low systemic absorption [13]. Although there is experimental evidence that proves the advantageous benefits of nebulized rather than intravenous CMS to treat inoculation pneumonia caused by Gram-negative bacteria [14], clinical studies confirming such a profit of aerosolized CMS in HAP and VAP are lacking [15].

Recent guidelines published by the European Society of Clinical Microbiology and Infectious Diseases (ESCMID) recommend against the use of nebulized CMS [16]. Based on ESCMID recommendations, clinical practice should refrain from utilizing nebulized antibiotics due to the limited amount of credible evidence supporting their effectiveness and the significant likelihood of undervalued adverse events, especially respiratory complications. Hence, ESCMID guidelines are advised to avoid the utilization of nebulized antibiotics in clinical settings until more robust evidence becomes available. In contrast, other guidelines propose the administration of nebulized CMS [1,17]. A significant issue that needs to be clearly defined is the exact dose of nebulized CMS, as, in most guidelines, the optional dose is not depicted [1,16,17]. In 2014, EMA suggested the daily administration of 3–6 MIU nebulized CMS [18]. However, there has been an increased interest in high-dose nebulized CMS, which is defined as a daily nebulized dose of >6 million IU (MIU). Recent pharmacokinetic data on the administration of high-dose aerosolized CMS (9–15 MIU daily) has shown favorable results in terms of pharmacokinetics (PK) [19,20] and accumulative clinical data on high-dose nebulized CMS for the treatment of DTR pathogens in HAP and VAP are coming into sight [21,22].

A narrative review of relevant studies was conducted using the PubMed/MEDLINE, Scopus, and Web of Science databases (from January 2000 until April 2023). The keywords used alone or in combination were as follows: aerosolized, nebulized, inhaled, colistin, colistin methanesulfonate, colistimethate, difficult to treat, difficult-to-treat-resistance, multidrug resistant, high-dose, Gram-negative, VAP, HAP, and nosocomial pneumonia. Information regarding clinical effectiveness and safety issues of high-dose nebulized CMS was included. Full text and abstract screening, as well as review articles, were searched.

In this review, the latest data regarding pharmacokinetics, clinical studies, and toxicity of high-dose nebulized CMS are analyzed.

## 2. CMS—Formed Colistin Characteristics

Colistin, also known as polymyxin E, is a cationic, multicomponent lipopeptide consisting of a cyclic heptapeptide with a tripeptide side chain acylated at the N terminus by a fatty acid. It has been isolated from *Bacillus polymyxa var. colistinus* and consists of two major components: colistin A (polymyxins E1) and colistin B (polymyxins E2). Although CMS is the form administered parenterally, it undergoes conversion in vivo to formed colistin, which is responsible for antibacterial activity, and thus, CMS should be considered an inactive prodrug. Colistin acts mainly on the surface of bacterial cell membranes [23]. It is active against most Enterobacterales, including carbapenemase-producing stains, independently of the resistance mechanism (excluding *Proteus* spp., *Providencia* spp., and *Serratia* spp.), as well as against carbapenemase-producing *A. baumannii* and DTR *P. aeruginosa* [24]. In a global in vitro study conducted in 2016–2018, colistin had the lowest resistance rate among MBL-positive isolates, and, regarding *P. aeruginosa*, colistin was the most active drug [25].

When administrated intravenously, CMS is eliminated mainly by the kidneys (~70%), whereas colistin undergoes extensive renal tubular reabsorption and predominately has a non-renal route of elimination. However, urinary concentrations of colistin in patients with normal kidney function after administration of CMS can be relatively high due to conversion from CMS within the urinary tract (25–30% of CMS is converted to colistin) [26]. Regarding CMS aerosol delivery, 9% of the dose typically reached the epithelial lining fluid (ELF), with only 1.4% absorbed as colistin [27].

## 3. Potential Benefits of Nebulized CMS

Intravenous CMS use is limited by a low safety profile due to renal and neurotoxicity, particularly when high dosages are required to achieve pharmacokinetic/pharmacodynamic (PK/PD) targets, as in the case of nosocomial pneumonia that require sufficient lung penetration [28,29]. In fact, clinical and pharmacologic data reveal that IV CMS demonstrates limited efficacy against respiratory tract infections [30]. In contrast, nebulized CMS attains high lung tissue concentrations [31], enabling rapid bacterial killing [32]. Optimized nebulization techniques, which make use of specially designed ventilation circuits and appropriately adjusted respiratory settings, result in the desired lung tissue penetration, contributing to efficient pathogen eradication, as demonstrated in an animal study [33]. By targeting the delivery of colistin, nebulized administration increases antibiotic concentrations in the ELF, especially in high doses and regardless of concomitant IV administration [19]. Intravenous dosing recommendations may be inadequate for critically ill patients with HAP/VAP, mainly due to altered pharmacokinetic profiles and decreased ELF penetration [29]. This targeted delivery not only boosts treatment efficacy but also reduces systemic exposure and potential adverse events, such as nephrotoxicity. High-dose drug nebulization, such as 5 MIU of CMS every 8 h, allows for high tissue concentrations while maintaining plasma levels below nephrotoxic levels [19]. Intrapulmonary CMS partially diffuses into the systemic compartment, with only 17% of the nebulized dose being rapidly eliminated by the kidneys [34]. It becomes evident that this approach may contribute to reducing healthcare costs due to prolonged and complicated hospitalization, as well as reducing renal replacement therapy interventions and improving patient therapy tolerance overall.

Nebulized CMS can also be safely administered at high and very high doses [35], owing to its favorable PK/PD properties and excellent bronchial tolerability, as evidenced by its use as a monotherapy in patients without cystic fibrosis [36]. It is a concentration-dependent antibiotic with a post-antibiotic effect. High-dose nebulization can be especially beneficial for critically ill patients with VAP due to the presence of a high bronchial inoculum [34]. The infected compartments of the lung act as a CMS reservoir, where it undergoes slow hydrolysis into colistin, providing continuous bacterial killing. Therefore, high colistin concentrations are achieved in the affected lung regions, enhancing the effectiveness of antibiotic treatment [37]. Another crucial consideration when choosing a high-dose regimen is the potential emergence of hetero-resistance due to bacterial exposure to suboptimal colistin dosages [38]. Overall, the localized distribution of nebulized CMS is key to unlocking its potential in terms of both efficacy and safety. To maintain colistin’s efficacy and prevent the development of resistance, it is essential to implement optimized dosing and administration techniques for high-dose CMS.

## 4. Dosage and Administration

The optimal high dose of nebulized CMS has not currently been defined. However, the effectiveness and safety of high doses of up to 15 MIU aerosolized CMS per day have been evaluated [20,21,22]. The CMS pharmacokinetic and pharmacodynamic properties, the excellent bronchial tolerability [22], and the low systemic absorption and associated toxicity [20] suggest that doses as high as 15 MIU can be recommended for difficult-to-treat Gram-negative lower respiratory tract infections. The administration of high-dose aerosolized CMS results in the achievement of elevated lung tissue concentrations if the nebulization procedure is optimized [19], and it compensates for colistin loss due to extrapulmonary disposition. In addition, the inoculum effect of CMS [34] and its slow hydrolysis to colistin in the lung support the administration of high doses. Minimal systemic exposure of colistin after nebulization of high doses of CMS and slow hydrolysis of CMS into colistin in plasma result in low serum concentrations, regardless of nebulization dose [20].

According to CMS product characteristics, 1 MIU of CMS should be dissolved in 3 mL of normal saline solution; thus, a 15 mL solution is required to deliver a dose of 5 MIU. Since the volume of most nebulization chambers ranges between 6 and 10 mL, the nebulizer needs to be filled at least twice. This increases the nurse’s workload, prolongs nebulization time beyond 60 min, and carries the risk of incomplete drug administration. A recent study [39] demonstrated that a reduction of diluent volume to 6 mL for nebulization of dilution of 4 million IU resulted in shorter nebulization time. In addition, compared with the 12 mL solution, the stability of the 6 mL solution was increased, and the nebulization time was significantly shortened. Moreover, no modification was observed in aerosol particle characteristics and plasma and urine pharmacokinetics. Therefore, dilution of 5 MIU of CMS with less than 10 mL of normal saline is possible.

CMS and colistin are not stable in aqueous media [39,40]; thus, reconstitution should be performed just before nebulization. Fatal acute respiratory distress syndrome has been described in a patient with cystic fibrosis and chronic airway infection with *Pseudomonas aeruginosa* after administration of a premixed CMS nebulization that was 5 weeks old [41]. In this patient, conversion of the colistin prodrug to the biologically active form of colistin due to prolonged storage of the aqueous solution was considered the likely cause of death.

## 5. Nebulizers and Nebulization Technique

The deposition of aerosol in the infected lung is directly influenced by the size of aerosolized particles and the type of nebulizer. Particles > 5 μm tend to deposit in the ventilator circuit and large airways, and the optimal median mass aerodynamic diameter ranges between 0.5 and 3 μm. Nebulization of high-dose colistin can be performed via jet (JN), ultrasonic (USN), and vibrating mesh nebulizers (VMN) which can generate aerosol particles with a diameter of < 5 μm.

JN are the most commonly used devices in invasively ventilated patients [42]. Nebulization is performed with exposure of the antibiotic to a highly pressurized air or oxygen flow delivered intermittently or continuously. Delivery of the drugs is not constant. Advanced ventilators perform synchronized nebulization, in which a fraction of the inspiratory flow is used to power nebulization. Compared with USN or VMN, JN has been shown to have the lowest efficiency due to high residual volume, accumulation of the drug in the circuit, and loss of drug through the expiratory limb [42,43]. According to previous studies, the best position for drug delivery is in the inspiratory limb 15 cm from the ventilator [44,45]. In recent studies, JN has been used for the administration of high-dose nebulized colistin [46,47,48].

In USN, a piezoelectric quartz crystal generates aerosol through vibration [42,43]. The efficiency of drug delivery is better than JN. In a previous study comparing the efficiency of JN and US in intubated patients, it was demonstrated that pulmonary deposition as a percentage of initial nebulizer activity was significantly greater with USN [49]. The best position for optimized drug delivery appears to be on the inspiratory limb at approximately 15 cm from the Y piece [44]. The main disadvantage of USN is the increase in solution temperature by 10 to 15 °C after 5 min of nebulization, which may affect the stability of nebulized CMS [42,43]. However, limited data exist regarding the administration of high-dose nebulized colistin with USN [50].

VMN consists of a small drug reservoir placed above a dome-shaped aperture plate with more than 1000 funnel-shaped apertures attached to a piezoceramic element [42,43]. The vibration of the aperture plate pumps liquid through the apertures, where it is broken into fine particles between 3 and 5 μm in size. The superiority of VMN over JN for nebulized antibiotics has been demonstrated in both in vitro and in vivo studies [35,51], and nebulization with VMN is recommended to optimize drug delivery to the lung. VMN is more efficient than JN and USN due to lower residual drug volume [44], and it does not affect solution temperature and, thus, stability [42,43]. In invasively ventilated patients, continuous rather than inspiration-synchronized nebulization is preferable as the latter requires extensively prolonged nebulization time [35]. Continuous nebulization results in the delivery of highly concentrated aerosol to the lung due to the bolus effect [35]. In addition, VMN operation without an external gas source and maintenance of ventilation delivery parameters are major advantages for clinicians [43]. Administration of high-dose nebulized colistin is mainly performed by VMN, as observed in recent studies [19,20,21,22,52,53]. In order to optimize drug delivery, the best VMN position in the circuit is 15 cm from the Y-piece in the inspiratory limb [44,45,54]. A comparison of the nebulizer generator devices is presented in Table 1 [13,35,42,43,44,51].

Limiting inspiratory flow velocity is important because it reduces turbulence, aerosol impaction, and drug deposition in the circuits, thereby optimizing lung deposition. To achieve that goal, it is important to use specifically designed smooth angles and inner surface circuits [35,43]. In addition, a specific ventilator setting should be applied during nebulization: volume-controlled ventilation with constant inspiratory flow, inspiratory/expiratory ratio < 50%, tidal volume at 8 mL/kg, respiratory frequency 12–15 bpm, and minimum bias flow (2 L/min) [35,43]. A plateau end-inspiratory pause of 20% of the duty cycle and positive end-expiratory pressure of 5 to 10 cm H_2_O should be applied to promote alveolar deposition [35]. During nebulization, the administration of a short-acting sedative or the transient increase in sedation is important to avoid patient–ventilator desynchrony [35,43]. In addition, the heat and moisture exchanger should be removed and heated humidification interrupted to avoid hygroscopic growth and massive trapping of aerosolized particles [35,43]. The placement of a filter on the expiratory limb is necessary to protect the ventilator flow device, and a filter change should be performed after each nebulization to avoid obstruction.

## 6. Pharmacokinetics of Nebulized Colistimethate Sodium

There is significant uncertainty relating to the benefits, optimal dosage, and clinical efficacy of nebulized colistin, mainly due to a lack of accurate PK data. The scarcity of these data may be explained, in part, by the complexity of PK and physicochemical properties of the drug, an intravenous form of which has been shown to exhibit inadequate penetration into lung tissue [29,55]. Existing PK data of nebulized colistin in critically ill patients with VAP caused by resistant pathogens were derived, so far, from a handful of small cohort studies.

### 6.1. The Backbone Studies on PK of Nebulized Colistin

A first effort to describe colistin concentrations in ELF of 20 critically ill patients with ventilator-associated tracheobronchitis (VAT) caused by polymyxin-only susceptible GNB was conducted by Athanassa et al. [31] via a high-pressure liquid chromatography (HPLC)-based method, which provides a more accurate analysis of colistin and its prodrug CMS and concentrations than the previously used microbiological assays [56]. A dose of 1 MIU, dissolved in 3 mL of half-normal saline, was administrated for over 30 min every 8 h via a vibrating-mesh nebulizer, the theoretically standard of care device, which generates the preferential particle size of the drug, achieving high concentrations in lung parenchyma [57]. After performing mini bronchoalveolar lavage, authors evaluated formed colistin ELF concentrations. Median values were 6.7 (4.8–10.1), 3.9 (2.5–6.0), and 2.0 (1.0–3.8) mg/L at 1, 4, and 8 h, respectively, and fivefold higher than those in plasma. However, measured ELF concentrations of the drug were below the MIC of isolated pathogens 4 h after inhalation, indicating the sub-optional dosage of 1 MIU of nebulized CMS.

A more proper characterization of the nebulized colistin PK was reported in the study of Boisson et al. [27], who investigated ELF and plasma CMS and formed colistin concentrations in 12 patients with VAP after inhalation of 2 MIU of CMS dissolved in 10 mL of saline and nebulized for over 30 min, followed 8 h later by the same dose of IV CMS. According to the PK model applied in this study, the range of colistin concentrations after inhalation was 9.53 to 1137 mg/L in ELF, and 0.15 to 0.73 mg/L in plasma, indicating that measured CMS and colistin concentrations in ELF were 100- to 1000-fold higher than those in plasma. Intravenous administration did not achieve therapeutic levels in the infection site [27].

### 6.2. ELF Formed Colistin Concentrations—Overstepping the Boundaries of Low Doses

The study by Gkoufa et al. was the first to evaluate the PK of CMS and formed colistin in ELF after high doses of nebulized CMS (3 MIU and 5 MIU) [19]. In detail, the study population included 30 patients with VAP divided into three equal groups. Ten patients received concomitantly IV and nebulized CMS of 3 MIU administered in 30 min via a vibrating-mesh nebulizer, 10 patients received 3 MIU of nebulized CMS as monotherapy, and for another 10 patients, 5 MIU of nebulized CMS was administrated as monotherapy. The applied PK model predicted the concentrations of CMS and formed colistin in ELF over 24 h, as well as estimated the unbound fraction of formed colistin. After a dose of 3 and 5 MIU of CMS, predicted trough concentrations of formed colistin in ELF were 120.4 mg/L and 200.7 mg/L, respectively. These concentrations ranged from more than 100- to 600-fold higher than those in plasma and more than 100-fold higher than the median MIC (i.e., 1 mg/L) of isolated pathogens. Regarding the free ELF concentrations of formed colistin, values were approximately 1- to 10-fold higher than the median MIC (i.e., 1 mg/L); however, after evaluating IV CMS administration as monotherapy, the formed colistin concentration in ELF was predicted to be much lower (>10-fold) compared with the nebulized groups. Moreover, IV administration of CMS did not contribute significantly to ELF-formed colistin concentrations. No safety issues of higher doses of nebulized CMS were raised from this study.

A more recent PK study using a high dose of nebulized CMS (5 MIU) investigated ELF CMS and colistin concentrations in seven critically ill patients with VAP [22]. PK results reported that one hour after nebulization, the median colistin and CMS concentrations in ELF were 121.7 (40.1–143.1) mg/L and 1445.3 (236.2–1918.2) mg/L, respectively, while twelve hours after nebulization, the median colistin and CMS concentrations were 122.6 (43.3–130) mg/L and 522.3 (222.3–636.5) mg/L, respectively. Colistin concentrations were far above the median MIC (1 mg/L) of isolated *Acinetobacter baumannii*, a finding of great importance not only in effectively treating lung infections but also in preventing the risk of acquisition of colistin resistance.

### 6.3. Plasma-Formed Colistin Concentrations

Systemic PK of high doses of nebulized CMS was assessed by Benitez-Cano et al., who investigated plasma colistin concentrations in 27 patients with VAP or HAP, 15 receiving 3 MIU and 12 receiving 5 MIU of CMS, dissolved in 6 mL and in 10 mL of saline, respectively, and nebulized for 30 min [20]. In this study, two types of nebulizers were used, a vibrating-mesh nebulizer in 17 patients and a jet nebulizer in 10 patients. Median (IQR) quantifiable formed colistin concentrations in plasma at 1, 4, and 8 h after nebulization of 3 MIU and 5 MIU of CMS were below 0.20 mg/L and 0.24 mg/L, respectively. Even high doses of nebulized CMS were proved to achieve undetectable or very low plasma colistin concentrations (<1 mg/L), being at the same time safe and well-tolerated, as the reported concentrations were much lower than those potentially reported to cause nephrotoxicity (~2.5 mg/L) [58]. A major limitation of the study was the lack of intrapulmonary PK data on nebulized colistin.

Similarly, a previously reported study evaluated the PK of CMS and formed colistin in plasma, besides those in ELF, after high doses of nebulized CMS (3 MIU and 5 MIU) using a population PK approach [19]. Authors indicated that free plasma concentrations of formed colistin after nebulization were minimal and below 1 mg/L across 24 h for both dosing groups and lower than those defined to cause nephrotoxicity, a result consistent with previous studies [20,27].

### 6.4. The Quandary of the Optimal Nebulizer. PK Data Resolves the Dilemma

As mentioned above, the available nebulization devices present considerable differences regarding their way of function and particle generation, and their availability mainly guides the decision for type selection. Moreover, after reviewing the literature for studies using nebulized colistin, researchers will discover a knowledge gap regarding the preferred nebulizer device, which may achieve sufficiently small particles of the drug in order to reach the pulmonary alveoli. Although published data support the superiority of vibrating mesh over jet and ultrasonic nebulizers, mainly due to advantages related to their manufacturing characteristics, lung PK data comparing different devices are missing, while in the study of Benitez-Cano et al. plasma colistin concentrations were higher with the use of vibrating-mesh compared to jet nebulizers [20]. A recent study by Kyriakoudi et al. compared ELF and plasma PK data of patients with VAP receiving 2 MIU of CMS either via a vibrating mesh or a jet nebulizer [59]. The maximum colistin concentrations in ELF, obtained with a vibrating mesh nebulizer were 10.4 (4.7–22.6) mg/L, while maximum ELF colistin values obtained with a jet nebulizer were 7.4 (6.2–10.3) mg/L. Regarding the C_max_ and C_min_ plasma formed colistin concentrations for the VMN were 2.6 (2.0–3.5) mg/L and 0.2 (0.1–0.3) mg/L, respectively, whereas for the JN, 0.3 (0.3–1.6) and 0.1 (0.1–0.2) md/L accordingly. Thus, the authors concluded that both nebulizers led to comparable formed colistin concentrations in ELF, providing a valuable finding in the field, as clinicians could probably have a safe and reliable alternative, considering the availability of devices and consumables in every hospital.

The aforementioned data indicate that higher doses of nebulized colistin may achieve adequate concentrations in lung compartments and, importantly, well above the MICs of isolated pathogens while, at the same time, eliminating systemic exposure and risk of nephrotoxicity and overcoming the obstacles of low penetration of intravenous colistin in ELF and adsorption of the drug to surfaces of sampling devices at low concentrations. However, looking across studies, plasma concentrations of formed colistin present variability after nebulization of different doses of CMS and do not follow a linear increase after administration of higher doses. Studies using low doses of nebulized CMS reported higher concentrations of colistin in plasma [31,59] compared with studies administrating high doses of nebulized CMS, which demonstrated low plasma colistin concentrations [19,20]. These discrepancies may be explained either by the PK properties of the drug or by the potential hydrolysis of CMS during analytic procedures, facts that may have biased measured colistin concentrations in plasma [60]. Notably, evaluating lung interstitial colistin concentrations in patients with VAP remains a challenge that requires a thorough investigation and approach and is still not fully elucidated. Factors that mainly hampered the achievement of therapeutic levels of this still potentially valuable antibiotic in the lung, and may influence measured concentrations, probably include the nebulized colistin dose and dosage interval, the possible contamination of the bronchoscope by bronchial secretions during the bronchoalveolar lavage, the adsorption of colistin to plastic surfaces, the methods used for the determination of colistin concentrations, and the population study—critically ill patients have different kinetics from other patients [61]. PK data of nebulized colistin should be further accompanied by observational studies based on clinical efficacy in order to reinforce its use by ensuring the beneficiary effect in infection-related outcomes.

## 7. Clinical Studies of High-Dose Nebulized CMS

The clinical benefit of nebulized CMS for the treatment of DTR Gram-negative hospital infections has been doubted, with conflicting data presented from different medical societies [1,16,17]. Most recommendations dealing with nebulized CMS are based mainly on the retrospective nature of many studies, the heterogenicity regarding the dosage scheme and the lack of optimization of the technique of nebulization, the inclusion of a small number of participants, as well as the lack of well-organized clinical trials to determine the rule efficacy of nebulized CMS in real-life conditions. Two different strategies of nebulized administration are considered clinically relevant and are classified as adjunctive and substitution. Adjunctive strategy is considered when nebulized CMS is concomitantly administered to patients already receiving IV CMS, added to standard first-line IV antibiotics. On the other hand, substitution strategy is defined as the use of nebulized CMS administered to patients not receiving IV CMS but only first-line IV antibiotics (other than colistin) [16].

It is of great significance to outline the current information regarding nebulized CMS. A meta-analysis of 12 studies, compromising 373 patients and including two randomized clinical studies, reported the effectiveness of nebulized colistin as monotherapy for respiratory tract infections due to MDR or Gram-negative susceptibility only to colistin with a clinical and microbiological success rate of 70% and a mortality rate of 33.8% [36]. Two meta-analyses demonstrated improved clinical and microbiological responses and lower infection-related mortality in patients receiving adjunctive treatment of nebulized colistin as a treatment for VAP and VAT caused by MDR Gram-negative bacteria compared to patients receiving intravenous therapy alone [62,63].

On the other hand, the largest meta-analysis emphasizing intravenous plus inhaled CMS versus intravenous CMS monotherapy, including 13 studies (11 retrospective and 2 prospective) and 1115 patients, concluded that no difference between each group was noticed in terms of mortality [15]. However, most studies included in the meta-analysis were up to 2016, where lower doses of nebulized CMS were administrated.

In the herein review, the clinical studies (from 2012–2023) focusing on the effectiveness of high-dose nebulized CMS are depicted in Table 2 [20,21,22,39,46,47,48,50,52,53]. High-dose nebulized CMS has been analyzed in nine studies and five prospective studies (including two pharmacokinetic studies with a clinical approach) [20,21,22,39,52], and the remaining were retrospective. High-dose nebulized CMS was administrated to approximately 670 patients with HAP and VAP caused by Gram-negative pathogens and mainly *A. baumannii*. Monotherapy with high-dose nebulized CMS was administered to 40% of patients, and a substitutive strategy was applied in 6 studies [20,21,39,46,48,52]. The median daily dose of nebulized CMS was 12 MIU (range: 6–15 MIU divided into 2–3 doses) with a clinical success of around 63% and mortality of 25%. However, in the era of the COVID-19 pandemic and with the increase in carbapenemase-producing *A. baumannii* infections [64], a higher mortality rate of 50.7% was observed [22].

The first study illustrating the beneficial use of high-dose nebulized CMS was reported by Lui et al. for the treatment of ventilator-associated pneumonia (VAP) caused by multidrug-resistant *Pseudomonas aeruginosa* and *Acinetobacter baumannii*. The study was prospective, observational, and comparative and included one arm with 122 patients with VAP caused by susceptible *P. aeruginosa* and *A. baumannii* and treated with intravenous β-lactam (ticarcillin/piperacillin, ceftazidime, or imipenem) for 14 days combined either with aminoglycoside (78% of patients) or quinolone (22% of patients) for 3 days. The second arm included 43 patients with VAP caused by multidrug-resistant *P. aeruginosa* and *A. baumannii* (sensitive only to colistin in 10 patients and resistant to all β-lactams but sensitive to colistin and aminoglycosides and/or ciprofloxacin in 33 patients) and treated with high-dose nebulized CMS of 5 MIU every 8 h either in monotherapy (*n* = 28) or combined to 3-day intravenous aminoglycosides (*n* = 15) for 7–19 days. Nebulization was performed with a vibrating plate nebulizer. All patients had received inappropriate initial antimicrobial therapy in the multidrug-resistant strain group, contrary to the sensitive strain group that received appropriate empirical treatment in 87% of cases. Clinical cure was similar between the two groups. In more detail, 29 of the 43 patients (67%) treated with nebulized colistin were characterized as clinically cured at the end of treatment compared with 81 of the 122 patients (66%) treated with intravenous β-lactams. Treatment failure with persistent VAP caused by *P. aeruginosa* was not statistically different between groups (*p* = 0.122), while the recurrence of VAP caused by *P. aeruginosa* and VAP caused by superinfection was analogous in both groups [21].

The effectiveness of high-dose nebulized CMS in VAP caused by MDR and XDR Gram-negative, mainly *A. baumannii* and *P. aeruginosa,* was evaluated in another prospective, randomized, single-blind trial conducted from 2013–2015. The study design comprised 73 patients with VAP treated with nebulized high-dose CMS and compared to 74 patients treated with intravenous CMS (a loading dose of 9 MIU followed by 4.5 MIU every 12 h was administrated). The nebulized CMS dose administrated in the nebulized camber was 7 MIU, and after extubating due to 40% pulmonary deposition, the calculated predicted dose in the respiratory tract was 4 MIU every 8 h, using an ultrasonic vibrating plate nebulizer. In the aerosolized group, patients were treated as monotherapy (*n* = 13) or in combination with intravenous β-lactam, mainly imipenem or tigecycline or quinolone (*n* = 53), whereas seven patients were excluded from the study. In the intravenous group, 12 patients were treated with intravenous CMS as monotherapy and 55 in combination, whereas 9 were excluded [52]. A clinical cure rate of 67% was observed in the nebulized group compared to 72% in the intravenous group, which was not statistically significant (*p* = 0.59). A favorable outcome regarding the nebulized CMS group was illustrated in terms of lower nephrotoxicity, prior weaning of ventilation, shorter time of bacterial eradication, and improvement in PaO_2_/FiO_2_ ratio during treatment. No difference was depicted in 28-mortality and length of stay. It is important to mention that in the aerosolized group, 2.7% of patients presented with bronchospasm [52].

The most recent prospective observational study conducted in 2020–2021, outlining the real-life clinical experience of 5 MIU of nebulized CMS every 8 h, using a vibrating mesh, comprised 134 ICU patients with VAP caused by *A. baumannii*, susceptible only to colistin. The baseline characteristics of the patients consisted of a SAPS II score of 43, with 43% presenting with septic shock and 25% complicated with secondary bacteremia. Aerolized CMS was administrated for a median duration of 10 days, and all patients were treated concomitantly with intravenous CMS (loading dose of 9 MIU, followed by 5.5 MIU every 12 h, whereas on continuous renal replacement therapy (CRRT) 6.75 MIU every 12 h) for a median duration of 8 days. Twenty-eight-day and ninety-day mortality were 50.7% and 58.2%, respectively, whereas clinical cure and microbiological eradication were 60.4% and 40.3%, accordingly. Nebulized high-dose CMS was not complicated by any adverse event. Multivariable analysis demonstrated lower SAPS II value, higher PaO_2_/FiO_2,_ and longer duration of nebulized CMS as independent factors of microbiological eradication [22].

The role of substitutive administration of nebulized high doses was initially evaluated in a retrospective study with 219 VAP infections caused by carbapenem-resistant *A. baumannii*. Nebulized CMS was administrated to 126 patients as monotherapy (*n* = 22) or with concomitant intravenous antimicrobial agents (*n* = 104), not including intravenous colistin, at a median daily dose of 9 MIU. The comparison group comprised 93 patients treated with intravenous antimicrobials, including intravenous colistin. The application of a propensity-score matched analysis revealed no significant differences in terms of clinical failure between the two groups; however, a major difference was observed regarding acute renal failure rates in favor of the nebulized group (18% versus 49%, *p* = 0.004). In addition, in the univariable analysis, despite the lack of statistical significance, a trend towards increased clinical failure was marked in patients treated with nebulized CMS at a daily dose below 9 MIU [46].

Another retrospective study focusing on the significance of the substitutive strategy of high-dose nebulized CMS included 557 cases of nosocomial pneumonia and VAP caused by Gram-negative pathogen susceptible to colistin and mainly *A. baumannii* (81.5%) and was conducted from 2016 to 2019. Substitutive nebulized CMS was administrated in 343 cases with intravenous antimicrobial agents other than intravenous CMS, mainly carbapenem and sulbactam. In the remaining cases, similar intravenous antimicrobial agents were prescribed without nebulized antibiotics. The baseline characteristics were HAP infections in 70.9%, with a median APACHE II score of 20 [IQR: 16–24] and a median SOFA score of 7 [IQR: 5–9], whereas septic shock was presented in 13% of cases. For dosage strategy analysis, nebulized CMS was categorized as low-dose nebulized CMS when the daily dose was ≤6 MIU CMS and high-dose nebulized CMS as >6 MIU. A propensity scoring (PS) matching was conducted and included 115 patient pairs with similar demographic characteristics in the final analysis. In the PS-matched cohort, clinical failure rates on days 7 and 28 were significantly lower in patients treated with substitutive nebulized CMS. Microbiological eradication was also significantly improved in the nebulized group. The all-cause mortality on day 28 was similar between the two groups in the PS-matched cohort as well as the ventilator weaning rates. The multivariable analysis of the PS-matched cohort revealed substitutive nebulized CMS as an independent factor of survival at day 14 and SOFA score as a predictor of clinical failure. Moreover, no difference was observed between low-dose and high-dose nebulized CMS in terms of clinical failure rates and mortality. However, a trend to lower clinical failure rates was identified in a subgroup of patients with age ≥75 years, smokers, chronic lung diseases, and an APACHE II score ≥20 that were treated with high-dose nebulized CMS and should be taken under consideration [48].

## 8. Toxicity—Adverse Events

Colistin-associated toxicity has been well established. The use of nebulized CMS results in high levels of the active compound, colistin in the broncho-alveolar tissue, limiting systemic diffusion, hence the risk of harmful systemic effects [20]. Nonetheless, the major side effects that remain of concern from the administration of inhaled colistin are nephrotoxicity, neurotoxicity, and bronchoconstriction.

CMS is well known for its nephrotoxicity, its most frequent side effect. Colistin tends to build up in the renal tubular cells, where it is reabsorbed via transporters causing high intratubular colistin levels and resulting in mitochondrial damage, cell apoptosis, and cell cycle arrest [37]. Furthermore, cell swelling and lysis caused by increased epithelial membrane permeability and oxidative stress and inflammation pathways are mechanisms that contribute to the development of colistin-induced kidney damage [65]. Nebulized CMS appears to be associated with reduced rates of renal impairment, nephrotoxicity, and need for renal replacement therapy when used as monotherapy compared to intravenous administration, as confirmed by multiple observational studies [46,52,53]. The rates of nephrotoxicity were also not increased in several studies where inhaled colistin was used as adjunctive therapy to iv administration [22,52,53]. When directly compared to β-lactams, no difference was reported in the incidence of renal toxicities [21]. Even when administered at higher doses, the concentration of colistin in the plasma remained at lower levels than those considered nephrotoxic (2.5 mg/L), which raised the possibility that the nephrotoxicity observed in some patients is probably not related to this treatment and associated with other factors, including hypovolemia, shock, disease severity, baseline renal function, etc.; or administration of other nephrotoxic compounds, including its intravenous form or other antibiotics [20,22,46]. This comes in line with a recent study indicating that low vs. high-dose nebulized inhaled colistin had comparable treatment outcomes and nephrotoxicity risk [48].

Neurotoxicity, which may be due to colistin administration, is a less common but still frequently observed dose-dependent side effect. It clinically presents with a variety of neurological symptoms, including paresthesia, neuromuscular blockade or apnea, etc., reflecting cell damage and neuronal cell death. Oxidative stress and mitochondrial damage have also been proposed as pathophysiological mechanisms [66]. Several cases of neurotoxicity from both intravenous and inhaled CMS administration have been reported so far; however, they seem to be associated with the co-presence of risk factors, including hypoxemia, co-administered muscle relaxant, narcotics, sedatives, or steroids, tending to increase the likelihood of its occurrence [52].

Bronchoconstriction is a side effect that seems to occur more frequently in patients using inhaled antibiotics as an adjunctive treatment [67]. The use of specific solutions with high osmolality and containing preservatives appears to be a reason behind these side effects [52]. Cases of administration of a compounded colistin solution, where the conversion of CMS to its active and toxic form had occurred prior to use, have been linked to direct lung toxicity [13]. The occurrence of bronchospasm is reported to be higher in patients with a history of hypersensitivity, such as asthma or obstructive pulmonary disease, or cystic fibrosis [13,68]. Although in some studies, no cases of bronchospasm were observed in patients treated with inhaled colistin due to the frequent use of bronchodilators, we cannot exclude the contribution of the latter in limiting the occurrence of bronchoconstriction [20]. Nevertheless, cases of bronchospasm noted in patients under treatment with inhaled colistin were successfully treated with the administration of bronchodilators [20]. Adverse events and toxicity of high-dose nebulized CMS are illustrated in Table 2.

It has also been reported that long-term use of IV colistin predisposes the selection of drug-resistant mutants due to poor lung penetration. [69,70]. Even though, theoretically, nebulized colistin reaching higher concentrations in infected lungs prevents selections of resistant pathogens [21], possible incomplete destruction of the bronchial epithelium by nebulization in combination with biofilm formation could facilitate colistin resistance and increase in MICs [71]. However, a low incidence of emergence of resistant strains has been reported [52,72,73].

## 9. Conclusions

In conclusion, nebulized CMS administration offers advantages in terms of efficacy against DTR Gram-negative bacteria, a favorable safety profile, and improved pharmacokinetics due to localized administration. The data indicate that higher doses of nebulized colistin may achieve adequate concentrations in lung compartments and, importantly, well above the MICs of isolated pathogens while, at the same time, eliminating systemic exposure and risk of nephrotoxicity and overcoming the obstacles of low penetration of intravenous colistin in ELF. The administration of high-dose nebulized CMS, defined as a nebulized CMS dose of above 6 MIU daily, specifically 3–5 MIU every 8 h, has been associated with a favorable clinical outcome and high microbiological eradication, whereas toxicities issues are limited. As a result, nebulized colistin holds promise as a valuable treatment option for nosocomial pneumonia, including HAP and VAP caused by resistant GNBs. However, the potential benefits of high-dose nebulized CMS warrant further in-depth examination. Nonetheless, further large-scale clinical trials and guidelines are necessary to optimize dosing regimens and nebulization practices for nebulized colistin to ensure safety and efficacy in the treatment of HAP and VAP.

## Figures and Tables

**Table 1 microorganisms-11-01459-t001:** Comparison of aerosol generator devices [13,35,42,43,44,51].

Nebulizer	Advantages	Drawbacks	% Dose Delivered
Jet nebulizer	Low cost Easy to use Small dimension Single use Breath synchronized	Low drug delivery Variable efficiency Interposition of gas/ventilator flow Long duration of nebulization Non-homogenous particle size	1–15
Ultrasonic nebulizer	Sustainable drug delivery Small interference of nebulizer/ventilator flow Quick drug delivery	Non-homogenous particle size Increase in solution temperature effect on stability Large dimension High cost Multiple use-hygiene concerns	30–40
Vibrating mesh nebulizer	Efficient drug delivery Homogenous particle size Low residual volume Synchrony with ventilator Constant solution temperature Low dimension Maintenance of ventilator settings	High cost Not suitable for concentrated and viscous solutions	40–60

**Table 2 microorganisms-11-01459-t002:** Clinical studies with high-dose nebulized CMS.

Author, Date	Type of Study	Subjects’ Neb Group	Control iv Group	Type of Infection/ Pathogen	Type of Nebulizer	Dose of Neb CMS/ Duration (Days, Range)	Dose of iv CMS/Duration (Days, Range)	N, Concomitant Antibiotics	Toxicity	Outcome	Comments
Lu et al., 2012 [21]	PCS, clinical efficacy	43 pts, neb mono: 28 neb + 3-day iv aminoglycoside: 15	122 pts, iv β-lactam + 3-day iv aminoglycoside	VAP/ 145, PA 20, AB (MDR and susceptible strains)	vibrating plate nebulizer	5 MU q8h/ 12 (7–19)	-	137, Aminoglycoside 122, β-lactam	No increase of risk of AKI with neb CMS	29/43 (67%) vs. 81/122 (66%) clinical cure. 19/28 of neb mono and 10/15 of neb + 3-day iv aminoglycoside clinical cure	Low risk of CMS resistance after neb. Neb CMS effective for MDR-GNB VAP with non-inferior clinical cure rate to that of VAP caused by susceptible GNB. Similar all-cause ICU mortality between groups.
Abdellatif et al., 2016 [52]	PCS, clinical efficacy	73 pts, neb mono:13	76, iv mono: 12	VAP/ AB, PA, KP (number of pathogens not defined)	ultrasonic vibrating plate nebulizer	4 MU q8h/ At least 14	LD 9 MU, MD 4.5 MU q12h/ At least 14	69, β-lactams 23, Aminoglycosides 11, Quinolones/macrolides 16, Tigecycline 14, Glycopeptides	Lower incidence of nephrotoxicity in neb vs. iv group (17.8 vs. 39.4%, *p* = 0.004) Moderate bronchospasm in 2.7% in the neb group	67.1% clinical cure rate in neb group and 72% in iv group, *p* = 0.59. TBE: neb vs. iv group 9.89 ± 2.7 vs. 11.26 ± 3 days, *p* = 0.023. Improvement of P/F ratio 349 vs. 316 at day 14, *p* = 0.012.	After extubating, 7 MU of neb CMS in the nebulizer chamber. No difference in the length of stay and the 28-day mortality
Jang et al., 2017 [53]	RCS, clinical efficacy	51	44	VAP/ AB	vibrating plate nebulizer	4.5 MU q8h/ 11.8 ± 5.4	LD: 9 MU MD: 4.5 MU q12h/ 10.9 ± 4.5	16, iv vancomycin 41, iv teicoplanin	Higher nephrotoxicity in the iv vs. neb group (60.5% vs. 15.7%, *p* < 0.0001)	79.6% clinical cure/improvement in the iv group vs. 76.5 in the neb group--65% microbiological eradication in the iv group vs. 66% in the neb group. Μortality rate 13.6% in the iv group vs. 19.6% in the neb group, *p* = 0.15.	Susceptibility to colistin using BMD method. Both groups had similar clinical and microbiological outcomes
Kim et al., 2017 [46]	RCS, clinical efficacy, PSM	126, neb mono: 22	93, iv mono: 36	VAP/ CR-AB	jet nebulizer	2.25 MU q12h to 4.5 MU q8 (median dose: 9 MU)/ 17 (10–25)	No LD, median daily dose 7.5 MU (4.5–9)/ 10 (7–16)	102, Carbapenems 25, Tigecycline 34, Minocycline 15, Ampicillin/sulbactam 45, Amikacin	AKI significantly more common in the iv vs. neb group (38% vs. 16%; *p* < 0.001) No cases of bronchospasm	57% clinical failure in the iv group vs. 39% in the neb group, *p* = 0.008. 59% ICU mortality in the iv group vs. 40% in the neb group, *p* = 0.006.	Susceptibility to colistin using BMD method. Neb CMS, without iv, effective and safe.
Bihan et al., 2018 [39]	PCS, PK, clinical efficacy	8	None	VAP/ PA, AB	vibrating-mesh nebulizer	4 MU q8h/ 9 (8–11)	-	-	NA	63% clinical cure rate 13% ICU mortality	6 mL of saline the preferred diluent volume over 12 mL due to shorter nebulization time, improved colistin stability, optimal particle size with no influence on plasma PK
Benitez-Cano et al., 2019 [20]	PCS, PK, clinical efficacy	27	None	21, HAP 6, VAP/ PA, ESBL Enterobacteriaceae	vibrating-mesh nebulizer in 17 patients jet nebulizer in 10 patients	3 MU q8h/ 7 (5–11) 5 MU q8h/ 5 (4–6)	-	-	No cases of neurotoxicity or bronchospasm AKI in six patients who were receiving other nephrotoxic drugs	19/27 (70%) clinical cure. 8/27 (29.6%) 30-day all-cause mortality.	Higher colistin concentrations with vibrating-mesh vs. jet nebulizer. Minimal systemic exposure and good tolerability of high doses of CMS.
Choe et al., 2019 [50]	RCS, clinical efficacy	35 (neb + LD iv)	156 (-non- LD iv: 70, -LD iv: 86)	140, VAP 51, HAP/ AB, PA, KP	ultrasonic nebulizer for intubated patients jet nebulizer for extubated patients	4.5 MU q8h/ 12 (6–16)	Median daily dose (mg/kg/day) in the non-LD iv group: 2.9 (2.1–4.3)/ 14 (10–15), in the LD iv group: 3.9 (2.9–5)/ 14 (9–15) in the neb LD iv group: 3.1 (2.2–4.1)/ 14 (12–17)	64, Carbapenem 21, Piperacillin/tazobactam 10, Minocycline 9, Tigecycline	No significant differences in nephrotoxicity between the non-LD iv group and the LD iv group Neb colistin did not increase the risk of nephrotoxicity	49% clinical cure rate in the neb–LD group vs. 46% in the non-LD iv vs. 42% in the LD iv group, *p* = 0.76–60% rate of microbiological eradication in the neb–LD group vs. 31% in the non-LD iv vs. 33% in the LD iv group, *p* = 0.010	No difference in clinical response between the three groups. Neb–LD group was significantly associated with lower mortality (adjusted OR 0.338, CI 95% 0.132–0.864, *p* = 0.024)
Casarotta et al., 2022 [47]	RCS, clinical efficacy	10 (iv CMS+ iv tigecycline +iv ampicillin/sulbactam + neb CMS)	22 (neb+ iv colistin alone or combined with another antibiotic)	Respiratory/ PDR AB	NA	3 MU q6h/ NA	LD 9 MU, MD 4.5 MU q12h/ NA	10, Tigecycline 10, Ampicillin/sulbactam 8, fosfomycin NA antibiotics of control group	40% (95% CI: [12, 73]%) AKI in the protocol vs. 4.5% (95% CI: [0.1, 22]%) in the control group, *p* = 0.01	100% vs. 36.4% microbiological negativization in the protocol vs. control group, *p* < 0.01. 100% (95% CI: [69, 100]%) vs. 36.4% (95% CI: [17, 59]%) survival from ICU in the protocol vs. control group, *p* < 0.01.	Susceptibility to colistin using BMD method
Feng et al., 2023 [48]	RCS, clinical efficacy	343, 165 > 6 MU q24h 178 ≤ 6 MU q24h	214, did not receive any form of CMS	395, HAP 162, VAP/ 454, CR-AB 48, CRE 55, CR-PA	jet nebulizer	2 MU to 15 MU q12h or 8h/ 7 (6–14)	-	276, Carbapenem 196, Sulbactam 156, Tigecycline	Similar dialysis rates with and without neb CMS	Clinical failure rate on days 7, 14, and 28, with and without substitutive neb CMS: 22.6% vs. 42.6%, *p* = 0.001, 27% vs. 42.6%, *p* = 0.013, and 27.8% vs. 41.7%, *p* = 0.027, respectively	High-dose neb CMS defined as >6 MU of CMS. Susceptibility to colistin using BMD method. No differences in clinical failure and mortality rates in patients receiving high and low dose of neb CMS. Microbiological eradication rates on day 14, and 28 significantly higher in patients with neb CMS
De Pascale et al., 2023 [22]	PCS, clinical efficacy, and PK	134 (neb + iv)	None	VAP/ COS-AB	vibrating mesh nebulizer	5 MU q8h/ 10 (5–13)	LD 9 MU, MD 5.5 MU q12h, 6.75 MU q12h during CRRT/ 8 (3–11)	25% of patients received cefiderocol, or fosfomycin	No drug-related adverse events	60.4% clinical cure from VAP. 40.3% microbiological eradication. 28- and 90-day mortality rates of 50.7% and 58.2%, respectively	High ELF concentrations in almost all samples at 1 and 12 h after neb delivery

Abbreviations: AB, *Acinetobacter baumannii*; AKI, acute kidney injury; BMD, broth microdilution; CMS, colistimethate sodium; COS, colistin only susceptible; CR, carbapenem resistant; CRE, carbapenem-resistant Enterobacterales; CRRT, continuous renal replacement therapy; ELF, epithelial lining fluid; ESBL, extended-spectrum beta-lactamase; GNB, Gram-negative bacteria; HAP, hospital-acquired pneumonia; Iv, intravenous; KP, *Klebsiella pneumoniae*; LD, loading dose; MD, maintenance dose; MDR, multi-drug resistant; Mono, monotherapy; MU, million units; NA, not applicable; Neb, nebulized; PA, *Pseudomonas aeruginosa*; PCS, prospective cohort study; PK, pharmacokinetics; RCS, retrospective cohort study; TBE, time to bacterial eradication; VAP, ventilator-associated pneumonia. Data are presented as mean (±standard deviation) or median (interquartile range).

## Data Availability

No new data were created or analyzed in this study. Data sharing is not applicable to this article.

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
