# Peer review of "High-Dose Nebulized Colistin Methanesulfonate and the Role in Hospital-Acquired Pneumonia Caused by Gram-Negative Bacteria with Difficult-to-Treat Resistance: A Review"

_microorganisms, 2023, doi:10.3390/microorganisms11061459_

Round 1
Reviewer 1 Report
The review "High-dose nebulized colistin methanesulfonate. Role in Gram-negative with difficult-to-treat resistance" highlights the significant burden of hospital-acquired pneumonia and the challenges associated with treating these infections. The authors suggest that high-dose nebulized CMS may have the potential to serve as an adjunctive or substitutive strategy for treating DTR Gram-negative bacteria, owing to its favorable safety profile and improved pharmacokinetics. The authors also provide a summary of the available evidence on high-dose nebulized CMS, including clinical studies, toxicity issues, and types of nebulizers.
While this review is significant in the field, it is crucial that the information and evidence are well-structured and well-written. Therefore, I believe it is appropriate for publication in Microorganisms, and I have no comments or suggestions regarding its publication.
Author Response
We would like to express our sincere gratitude for your invaluable review of our manuscript. We genuinely appreciate the time and effort you have dedicated to evaluating our research.
Reviewer 2 Report
“High-dose nebulized colistin methanesulfonate. Role in Gram-negative with difficult-to-treat resistance; a review”
By Karaiskos et al.
There is something wrong in this submission. In the manuscript that I am downloading, there is no table but there are mentions of two tables. I don’t know whether the problem is with my computer or the publisher’s website or in the authors’ submission. However, since the results have been well described in the text, I assume that the tables have also been well constructed.
In this review the authors describe the use of nebulized colistin methanesulfonate (CMS, a prodrug of colistin), a method that provides colistin at a concentration higher than MIC at the infection site but results in a lower concentration in blood than what is required for causing nephrotoxicity. It is not clear from the review, how commonly used is nebulized CMS clinically. Since the ESCMID recommends against its use, I think the authors should discuss a little more about the opinion of the European Society and why they recommend against the method. Since the manuscript is a review of the topic, both sides should be presented. I also have the following minor comments on the manuscript.
Line 2-3: I have never seen a title written this way. Why is it written in quotation mark (“”)? Why is there a period in the middle of the title? Is the first line supposed to be a sentence without a verb? A title need not have a verb, unless it is written as a sentence. “Gram-negative” is an adjective but here it is not followed by a noun (such as bacteria). The purpose of the semicolon (;) is also not clear.
Page 1 Line 34: Change “pandemic” to “pandemic era”
Page 2 Line 50: Change “b-lactamase” to “b-lactamase” with the b in symbol font
Page 2 Line 54: ESBL, KPC, MDR, XDR and others. Please provide a list of abbreviations or write the full form when introduced for the first time.
Page 2 Lines 69-71: “Recent guidelines……nebulized CMS”. Although references have been provided, for the benefit of the reader, is it possible to briefly state the main reasons for the recommendations against the use of nebulized CMS?
Page 3 Line 101: “MBL” This is the only time this abbreviation has been used in the manuscript and it has not been explained what the abbreviation stands for.
Page 3 Line 106: The explanation given in parenthesis “(…)” should be placed at the end of the sentence.
Page 3 Line 108: “ELF” Since this abbreviation has been used here for the first time, please mention what this stands for. That has actually been done in Line 121.
Page 3 Line 123: Delete comma “,”.
Page 4 Line 156: Change “and its slow hydrolysis of CMS in colistin” to “and its slow hydrolysis to colistin”
Page 4 Line 163, 171, 180, 184, 192 and other places: Be consistent about adding or not adding a space before ml, mm, cm etc.
Page 4 Line 172-177: This paragraph raises several questions. CMS is hydrolyzed to colistin. What are the breakdown product(s) of colistin? Is it known whether it is colistin or its breakdown products that are responsible for the fatal effect? Has the rate of breakdown of CMS and/or colistin in vitro or in vivo been studied?
Page 4 Line 193: Add space before “In”
Page 4 Line 200: Change “temperature solution” to “solution temperature”
Page 5 Line 224: Add space after “circuits”
Page 5 Line 247: Not clear why et al. is always followed by a semicolon “;” instead of a comma “,”. Is that a journal requirement?
Page 5 Line 249: Change “previous used” to “previously used”
Page 5 Line 250: “administrated over 30 minutes and every 8 hours” Not clear what this means. I think, it should be changed to “administered for over 30 minutes every 8 hours”
Page 6 Line 262: Change “nebulized over” to “nebulized for over”
Page 7 Line 323: Change “overjet” to “over jet”
Page 9 Line 440: Change “a percentage of 2.7%” to “2.7%”
Quality of English is fine. There are a few minor typographical errors.
Author Response
Detailed response to review
There is something wrong in this submission. In the manuscript that I am downloading, there is no table but there are mentions of two tables. I don’t know whether the problem is with my computer or the publisher’s website or in the authors’ submission. However, since the results have been well described in the text, I assume that the tables have also been well constructed.
In this review the authors describe the use of nebulized colistin methanesulfonate (CMS, a prodrug of colistin), a method that provides colistin at a concentration higher than MIC at the infection site but results in a lower concentration in blood than what is required for causing nephrotoxicity. It is not clear from the review, how commonly used is nebulized CMS clinically. Since the ESCMID recommends against its use, I think the authors should discuss a little more about the opinion of the European Society and why they recommend against the method. Since the manuscript is a review of the topic, both sides should be presented. I also have the following minor comments on the manuscript.
Response: Thank you for your valuable comment. A sentence has been added to analyse ESCMID suggestions on nebulized antibiotics.
Line 2-3: I have never seen a title written this way. Why is it written in quotation mark (“”)? Why is there a period in the middle of the title? Is the first line supposed to be a sentence without a verb? A title need not have a verb, unless it is written as a sentence. “Gram-negative” is an adjective but here it is not followed by a noun (such as bacteria). The purpose of the semicolon (;) is also not clear.
Response: The title has been modified to accommodate to review suggestion.
Page 1 Line 34: Change “pandemic” to “pandemic era”
Response: Correction has been made
Page 2 Line 50: Change “b-lactamase” to “b-lactamase” with the b in symbol font
Response: Correction has been made
Page 2 Line 54: ESBL, KPC, MDR, XDR and others. Please provide a list of abbreviations or write the full form when introduced for the first time.
Response: Abbreviations have been described in the manuscript
Page 2 Lines 69-71: “Recent guidelines……nebulized CMS”. Although references have been provided, for the benefit of the reader, is it possible to briefly state the main reasons for the recommendations against the use of nebulized CMS?
Response: A comment has been added to accommodate to review suggestion
Page 3 Line 101: “MBL” This is the only time this abbreviation has been used in the manuscript and it has not been explained what the abbreviation stands for.
Response: Modification has been made to accommodate to review suggestion.
Page 3 Line 106: The explanation given in parenthesis “(…)” should be placed at the end of the sentence.
Response: Modification has been made to accommodate to review suggestion.
Page 3 Line 108: “ELF” Since this abbreviation has been used here for the firs time, please mention what this stands for. That has actually been done in Line 121.
Response: Modification has been made to accommodate to review suggestion.
Page 3 Line 123: Delete comma “,”.
Response: Modification has been made to accommodate to review suggestion.
Page 4 Line 156: Change “and its slow hydrolysis of CMS in colistin” to “and its slow hydrolysis to colistin”
Response: Modification has been made to accommodate to review suggestion.
Page 4 Line 163, 171, 180, 184, 192 and other places: Be consistent about adding or not adding a space before ml, mm, cm etc.
Response: Modification has been made to accommodate to review suggestion.
Page 4 Line 172-177: This paragraph raises several questions. CMS is hydrolyzed to colistin. What are the breakdown product(s) of colistin? Is it known whether it is colistin or its breakdown products that are responsible for the fatal effect? Has the rate of breakdown of CMS and/or colistin in vitro or in vivo been studied?
Response: CMS is an inactive drug that is hydrolysis to a complex mixture of many different sulfomethyl derivatives, including colistin A and B, that are the active compounds and enhance antibacterial effect. Based on the report, colistin A was the most possible etiological agent. Inhalation toxicologic studies in rats and dogs showed massive inflammation at low doses of colistin in lungs. The rate of conversion has been studied. For your reference, 25–30% of CMS is converted to colistin in vivo. Therefore, CMS should be reconstituted just before administration to avoid excessive conversion to active colistin, which causes airway or alveolar damage. These details are analysed in the manuscript.
Page 4 Line 193: Add space before “In”
Response: Modification has been made to accommodate to review suggestion
Page 4 Line 200: Change “temperature solution” to “solution temperature”
Response: Modification has been made to accommodate to review suggestion
Page 5 Line 224: Add space after “circuits”
Response: Modification has been made to accommodate to review suggestion.
Page 5 Line 247: Not clear why et al. is always followed by a semicolon “;” instead of a comma “,”. Is that a journal requirement?
Response: This is based on journals requirement.
Page 5 Line 249: Change “previous used” to “previously used”
Response: Modification has been made to accommodate to review suggestion
Page 5 Line 250: “administrated over 30 minutes and every 8 hours” Not clear what this means. I think, it should be changed to “administered for over 30 minutes every 8 hours”
Response: Modification has been made to accommodate to review suggestion
Page 6 Line 262: Change “nebulized over” to “nebulized for over”
Response: Modification has been made to accommodate to review suggestion
Page 7 Line 323: Change “overjet” to “over jet”
Response: Modification has been made to accommodate to review suggestion
Page 9 Line 440: Change “a percentage of 2.7%” to “2.7%”
Response: Modification has been made to accommodate to review suggestion